# Decoding immunometabolism with next-generation tools: lessons from dendritic cells and T cells

Yi-Hao Wang [1,2,4], Limei Wang [1,2,4 ✉] & Ping-Chih Ho [1,2,3 ✉]

## Abstract

**Cellular metabolism plays a pivotal role in regulating the effector functions and fate decisions of immune cells, shaping immune responses in homeostasis and disease. Metabolic pathways also serve as critical signaling hubs governing immune cell behavior. Deregulated metabolic pathways contribute to immune dysfunction, fueling disease progression and creating challenges for therapeutic interventions. The recent development of advanced technologies to delineate immunometabolic regulation has revolutionized our understanding of immune cell biology. These tools, ranging from quantitative single-cell metabolomics to in vivo spatial tissue profiling and DC-based metabolic therapy, have shifted the focus from broad nutrient pathways to a detailed exploration of metabolic reprogramming within disease microenvironments, revealing how metabolic changes drive immune cell activation, differentiation, and effector responses. The integration of immunometabolic insights into clinical practice holds strong potential for advancing precision medicine and developing targeted therapies that restore immune balance in pathological conditions. Here, we summarize emerging cutting-edge technologies related to immunometabolism and critically reflect on their current limitations. Finally, we discuss potential needs for developing novel methods that can uncover the intricate interplay between metabolism and immune cell function.**

**Keywords** Immunometabolism; Metabolic Reprogramming; Dendritic Cells; T Cells; Technological Advances
**Subject Categories** Immunology; Metabolism

## Introduction

Over the past two decades, immunometabolism has undergone a remarkable transformation, propelled by technological innovation, deeper insights into cellular metabolism, and the growing appreciation of its central role in orchestrating immune cell function. Advances in cutting-edge technologies have expanded the field beyond the study of intracellular signaling metabolites and

metabolic pathways, enabling multi-layered analyses that capture the complexity of metabolic-immune crosstalk. Immunometabolic regulation operates across multiple levels, from the dynamic actions of signaling metabolites to sophisticated analytical platforms that unravel the intricate interplay between metabolism and immune responses, offering fresh perspectives on immune regulation and therapeutic opportunities. In this review, we provide a concise overview of immunometabolic regulation in immune cells. Given the pivotal role of dendritic cells (DCs)-the most potent antigen-presenting cells-in orchestrating T cell activation and differentiation, and the ability of T cells to dynamically reprogram their metabolism in response to proliferative, migratory, and effector demands shaped by the tissue microenvironment, we focus primarily on DCs and T cells in both physiological and pathophysiological settings. We also highlight key challenges and bottlenecks in immunometabolism research, and discuss emerging tools and strategies designed to overcome these hurdles.

## Technological advances in tools to study the dynamic nature of immune cells and metabolic profiles in the microenvironment

### Technological advances in metabolite measurement and metabolic activity analysis

The evolution of cellular metabolism research began with bulk analysis methods, including extracellular flux analysis (EFA) and the introduction of the Seahorse Bioscience XFe Extracellular Flux Analyzer (Fig. 1). The advent of the Seahorse technology expanded investigations from broad upstream biology to real-time metabolic analysis of live cells, enabling mechanistic insights at cellular and molecular levels. These approaches have since been widely applied to study metabolic changes in various immune cells, including DCs and T cells (Møller et al, 2022; Peng et al, 2023). However, traditional bulk metabolism methods are constrained by several limitations, including a focus on steady-state metabolomics, reliance on in vitro assays, low throughput, and restricted coverage of metabolic pathways beyond glycolysis and the tricarboxylic acid (TCA) cycle (Wei et al, 2021). Advances in flow cytometry enabled metabolic profiling of rare cell populations and at single-cell resolution. For instance, the single-cell energetic metabolism *by profiling translation inhibition* (SCENITH) assay measures cellular

¹Department of Oncology, University of Lausanne, Lausanne, Switzerland. ²Ludwig Institute for Cancer Research, University of Lausanne, Épalinges, Switzerland. ³College of Medical Science and Technology, Taipei Medical University, Taipei, Taiwan. ⁴These authors contributed equally: Yi-Hao Wang, Limei Wang. ✉E-mail: limei.wang@unil.ch; ping-chih.ho@unil.ch

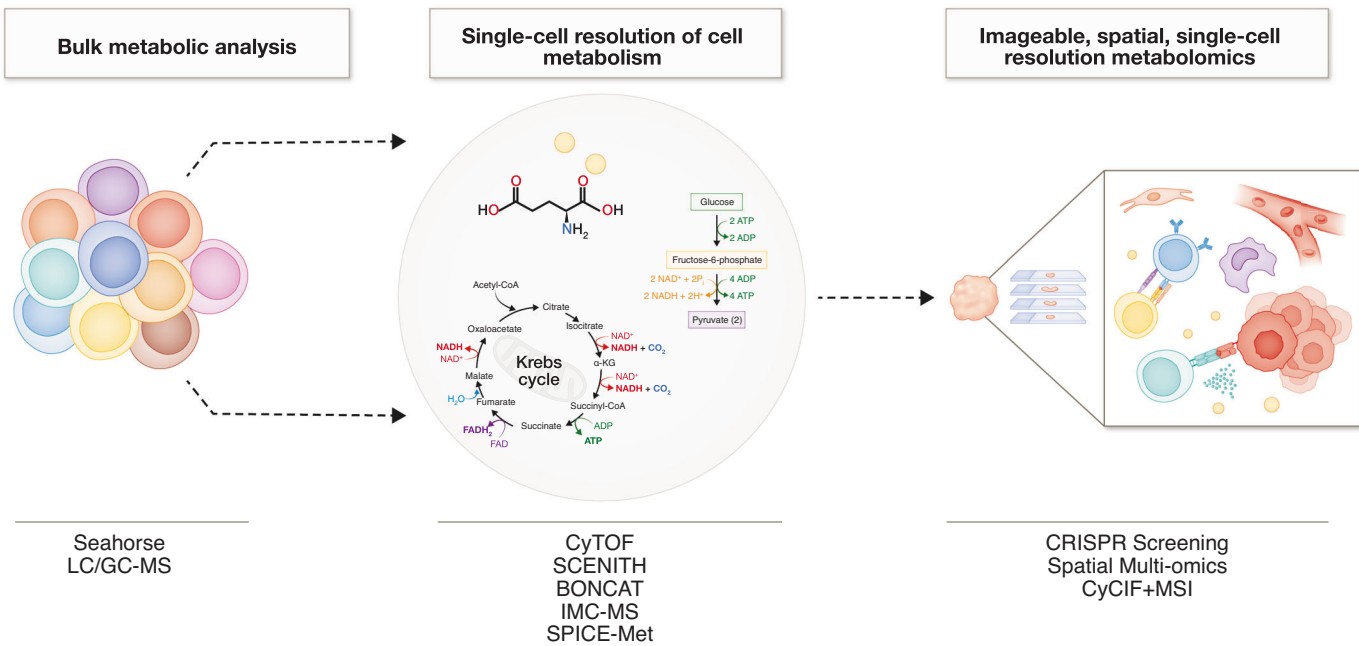

**Figure 1. The evolution of technologies in studying cell metabolism.**

The study of cell metabolism has evolved significantly over time. Initially, research focused on analyzing mixed cell populations and extracellular fluids to provide an overall view of cellular metabolomics. Techniques such as Seahorse analysis evaluated metabolic activity, while LS/GC-MS characterized metabolite composition. Advancements in flow cytometry and improved single-cell resolution technologies, such as CyTOF and SCENITH, have provided deeper insights into cellular metabolism at the single-cell level. These technologies not only enabled detailed analysis of specific cell types but also facilitated the study of rare cell populations. More recently, the integration of multi-omics approaches, single-cell technologies, imaging systems, and spatial information has opened new avenues. This combination allows for the investigation of cell metabolism within a spatial context, offering a comprehensive understanding of how metabolism correlates with cellular location in tissues and specific cell populations. Additionally, in vivo CRISPR screening has opened the possibility of identifying key metabolic genes and their roles in regulating cell metabolism at the single-cell level and within a spatial framework. The evolution of these technologies has pushed the field of metabolism research into an era of unprecedented detail, uncovering the complexity of cellular metabolism.

protein synthesis as a proxy for metabolic activity at the single-cell level, facilitating global metabolic profiling of heterogeneous samples, including blood, lymphoid tissues, and tumors (Argüello et al, 2020). Single-cell metabolic profiling (scMEP) is another method which has been used to reveal metabolic regulation in DCs (Munson et al, 2023). Coupling SCENITH with CyTOF-based metabolic regulome profiling identified mitochondrial dependence as a critical parameter for DC maturation and a predictive biomarker of clinical response. In melanoma, patients with superior overall survival (OS) and progression-free survival (PFS) exhibited enhanced mitochondrial metabolism, fatty acid oxidation (FAO), and glutaminolysis in matured DCs (Adamik et al, 2023). To circumvent the cytotoxic effects of puromycin used in SCENITH, newer methods such as bioorthogonal noncanonical amino acid tagging (BONCAT) employ noncanonical amino acids (ncAAs) to measure protein synthesis inhibition without toxicity (Vrieling et al, 2024). Nevertheless, cytometry-based approaches still face limitations, including dependence on antibody labeling, and ex vivo incubation makes SCENITH with indirect detection. To facilitate a clearer understanding of the strengths and limitations of these methodologies, Table 1 summarizes key features, advantages, and disadvantages of the main metabolic profiling approaches discussed.

Metabolomics has emerged as a powerful, high-throughput approach for investigating steady-state metabolite profiles using techniques such as liquid or gas chromatography-mass spectrometry (LC-MS or GC-MS) and nuclear magnetic resonance (NMR) (Fu et al, 2023). Recent innovations in single-cell metabolomics have extended its applications, providing a powerful platform for systems biology and immunology research (Hartmann et al, 2021). For instance, highly sensitive metabolomics combined with transcriptomics revealed that pregnenolone steroids associated with parasitemia exert immunosuppressive effects during *Plasmodium falciparum* infection (Abdrabou et al, 2021). Untargeted comparative metabolomics is particularly valuable for discovering a broad spectrum of metabolites, including microbiota-derived molecules that influence host biology. LC-MS/MS-based metabolomic screening identified the gut microbe–derived metabolite trimethylamine N-oxide (TMAO) as an enhancer of IFN signaling pathways, promoting an immunostimulatory phenotype in tumor-associated macrophages (TAMs) and stimulating effector T cell responses (Mirji et al, 2022). Similarly, plasma metabolomics of newborns vaccinated with Bacille Calmette-Guérin (BCG) revealed BCG-induced changes in lysophosphatidylcholines (LPCs), correlating with cytokine responses and suggesting LPCs as mediators of BCG immunogenicity (Diray-Arce et al, 2022). Enhancement of

**Table 1. Summary of methods for analyzing immune cell metabolism, highlighting their key applications, strengths, and limitations.**

| Technology/Method | Application | Advantages | Limitations |
|---|---|---|---|
| Extracellular flux analysis (Seahorse) | Measures OCR and ECAR to assess mitochondrial respiration and glycolysis | Real-time, live-cell analysis; widely used | Low throughput; bulk analysis only; limited pathway coverage |
| SCENITH | Single-cell metabolic activity via puromycin-based translation inhibition | Flow cytometry-based; single-cell resolution; compatible with immune tissues | Cytotoxicity from puromycin; indirect measurement; requires antibody panels |
| Single-cell metabolomics (scMEP) | Profiles metabolites in individual cells | High resolution; reveals cell-to-cell variability | Technically challenging; low throughput; data interpretation complexity |
| Bioorthogonal noncanonical amino acid tagging (BONCAT) | Non-toxic measurement of protein synthesis using noncanonical amino acids | Less toxic than SCENITH; compatible with live cells | Requires chemical labeling; limited temporal resolution |
| LC-MS / GC-MS | Quantitative/qualitative metabolomics; targeted or untargeted analysis | High sensitivity and coverage; broad metabolite detection | Requires high input; extensive sample prep; complex data analysis |
| IMC-MS (Inertial microfluidic chip-MS) | Metabolic profiling of single immune cells (e.g., macrophage polarization) | Combines microfluidics and MS; single-cell insights | Requires specialized equipment; technical complexity |
| SPICE-Met /Perceval HR | Imaging ATP:ADP ratios in live cells | Real-time spatial resolution; live-cell compatible | Requires fluorescent reporters; limited multiplexing |
| COMPASS | Predicts metabolic flux from scRNA-seq using flux balance analysis | Computational integration of metabolism and transcriptomics | Predictive model; requires high-quality scRNA-seq |
| iMetAct | Infers enzyme activity and metabolic network rewiring from transcriptomics | Identifies metabolic subtypes; accessible web interface | Requires validation; inference-based |
| CRISPR screens (Loss/gain of function) | Identify metabolic regulators in immune cells | Functional; genome-wide screening | Off-target effects; compensatory mechanisms; validation required |
| Spatial multi-omics (transcriptomics, proteomics, metabolomics) | Maps the spatial organization of immune and metabolic states in tissues | High-dimensional; tissue context preserved | High cost; integration complexity |

these shifts with high levels of cyclic di-AMP further promoted glycolysis and immunogenicity (Singh et al, 2022). However, despite these advances, many MS-based metabolomic methods still require high input flux and extensive computational analysis, limiting their scalability.

The advent of single-cell technologies has further expanded metabolomics applications in immunology. Integration of high-dimensional flow cytometry and single-cell metabolomics revealed a hypoxia-driven shift from mitochondrial respiration and FAO toward glucose-dependent anaerobic metabolism in immune cells (Gurshaney et al, 2023). Inertial microfluidic chip-mass spectrometry (IMC-MS) enabled identification of key metabolic drivers of macrophage polarization, notably glutamine promoting M1 polarization in the tumor microenvironment (Hu et al, 2024). Innovations such as single-cell profiling and imaging of cell energy metabolism (SPICE-Met), utilizing the Perceval HR fluorescent sensor to monitor intracellular ATP:ADP ratios, have provided unprecedented insights into the metabolic heterogeneity of complex immune cell populations (Russo et al, 2022). Computational frameworks have further propelled the field. COMPASS integrates single-cell RNA sequencing (scRNA-seq) with flux balance analysis to predict cell-type-specific metabolic fluxes, uncovering a metabolic switch between glycolysis and FAO that governs T helper 17 (Th17) cell pathogenicity (Wagner et al, 2021). Similarly, iMetAct infers enzyme activity from gene expression, integrating metabolic-transcriptional networks to identify metabolically distinct subtypes in hepatocellular carcinoma, and enabling single-cell analyses of tumor metabolism and immune interactions

via an accessible online platform (Wang et al, 2025). Refinement of in vivo CRISPR screening has revolutionized the identification of metabolic regulators of immune cell function. Pooled loss-of-function screens targeting metabolic genes identified nutrient signaling pathways, including *Slc7a1*, *Slc38a2*, and *Pofut1*, as critical determinants of CD8[+] T cell fate, with Pofut1 deficiency enhancing memory responses in tumor and viral models (Huang et al, 2021). A genome-scale, gain-of-function CRISPR screen further identified proline dehydrogenase 2 (PRODH2) as a key driver of metabolic reprogramming that enhances CAR-T cell efficacy (Ye et al, 2022). Furthermore, CRISPR-Cas9-based targeted screening also revealed methylenetetrahydrofolate dehydrogenase 2 (MTHFD2) as essential for sustaining purine synthesis in activated CD4[+] T cells (Sugiura et al, 2022b). Nevertheless, challenges persist: network complexity, compensatory mechanisms, and technical limitations such as off-target effects and incomplete gene disruption complicate interpretation, underscoring the need for complementary approaches and rigorous validation. Finally, spatially resolved multi-omics technologies have emerged as powerful tools to dissect the spatial heterogeneity of tissue microenvironments (Sun et al, 2023). Spatial genomics, transcriptomics, proteomics, and metabolomics provide a comprehensive map of cellular and molecular interactions (Du et al, 2024). In glioblastoma, integration of spatial transcriptomics, metabolomics, and proteomics revealed that oxidative stress drives chromosomal rearrangements, promoting clonal evolution (Ravi et al, 2022). Future advances combining spatial multi-omics with artificial intelligence (AI) promise three-dimensional reconstructions of the tumor microenvironment

(TME), paving the way for transformative improvements in cancer diagnosis and therapy.

## Metabolic environments in the initiation and progression of pathological conditions

Metabolic profiling, the comprehensive analysis of metabolites within biological systems such as fluids, cells, or tissues, has emerged as a critical tool for understanding immunometabolic regulation (Nicholson et al, 1999). Yet, the direct connection between metabolite signatures and specific pathological states remains incompletely understood. By integrating advanced detection platforms, researchers are beginning to unravel how metabolic environments shape immune responses and contribute to disease. Recent progress in this field has been driven by the application of cutting-edge technologies combined with rigorous experimental design, enabling the dissection of complex metabolic-immune interactions and the identification of novel therapeutic opportunities.

By applying those techniques, several findings have been uncovered in vivo. A recent work integrating cyclic immunofluorescence (CyCIF) with mass spectrometry imaging (MSI) uncovered how the immunosuppressive metabolite D-2-hydroxyglutarate (D-2HG) modulates lactate dehydrogenase (LDH) activity, thereby regulating CD8$^+$ T cell proliferation within the TME of human gliomas (Notarangelo et al, 2022). These findings highlight the immunomodulatory role of oncometabolites and demonstrate the power of multi-modal imaging to reveal critical metabolic-immune interactions in cancer (Wymann and Schneiter, 2008). Lipid metabolism is essential for membrane dynamics, signaling, and energy production, particularly in proliferating cancer cells. However, the precise lipid species that facilitate immune evasion remain poorly defined. CRISPR screens targeting lipid metabolism pathways revealed that de novo synthesis of sphingolipids, phosphatidylcholine, and phosphatidylethanolamine promotes tumor immune escape. Complementary lipidomic analyses, coupled with FACS-based genetic screens, identified glycosphingolipids as key regulators of IFNγ receptor subunit 1 (IFNGR1) surface expression, linking lipid metabolism to immune surveillance and immunotherapy responses (Soula et al, 2024). In autoimmune diseases such as type 1 diabetes (T1D), rheumatoid arthritis, and multiple sclerosis, dysregulated immunity arises from the interplay between genetic susceptibility and environmental triggers (Tsai et al, 2021). The etiology and pathogenesis of autoimmune diseases are likely triggered by the interplay of genetic predisposition and environmental risk factors, progressing through dysregulation of the immune system, including innate and adaptive immune cell infiltration into specific organs. Over the past decade, the gut microbiota has emerged as a key contributor to autoimmune diseases, primarily through the enhanced production of pro-inflammatory cytokines stimulated by commensal microbiota (Tsai et al, 2021). Intriguingly, large multicenter cohort studies have identified microbial metabolic pathways—particularly short-chain fatty acid (SCFA) synthesis and fermentation—as major discriminators between T1D patients and healthy controls (Vatanen et al, 2018). Metabolomic profiling using LC-MS/MS and GC-MS revealed that T1D patients exhibit increased lipopolysaccharide biosynthesis and reduced butyrate and bile acid metabolism, establishing a distinct immunometabolic signature (Yuan

et al, 2022). Beyond autoimmunity, metabolomic studies have shed light on metabolic dysregulation in infectious and neurodegenerative diseases. In dental infections, periapical abscesses exhibit enrichment of lipid metabolites such as 16-20-HETE and 17-octadecynoic acid compared to healthy pulp tissue (Altaie et al, 2021). In Alzheimer's disease (AD), microglial dysfunction is central to disease progression; multi-omics analyses revealed that exposure to fibrillar Aβ induces dysregulation of lipid metabolism, lysosomal impairment, and foam cell phenotypes, driving inflammatory responses (Xia et al, 2022). Similarly, Mycobacterium tuberculosis (Mtb) infection, especially with drug-resistant strains, induces profound immunometabolic alterations (Howard and Khader, 2020). T cells from Mtb-infected hosts display impaired mitochondrial metabolism, as shown by carbon tracing and metabolomics analyses (Russell et al, 2019). Moreover, live Mtb infection enhances IFN signaling, diminishes glycolysis, and exacerbates mitochondrial damage in macrophages, further illustrating the metabolic vulnerabilities during chronic infection (Olson et al, 2021). Together, the integration of advanced methodologies—including multi-omics profiling, metabolic flux analysis, and spatially resolved imaging—continues to transform our understanding of the dynamic relationship between metabolism and immune function. These efforts are accelerating the discovery of disease-associated metabolic regulators in homeostatic and pathological conditions and paving the way for targeted therapies and precision immunometabolic interventions.

# Cell type-specific immunometabolic regulations: lessons from dendritic cells and T cells

Technological advancements have been transformative, enabling unprecedented insights into DC and T cell biology. As a cornerstone of immune cell physiology, metabolic pathways intricately control activation thresholds, differentiation programs, and effector responses through dynamic regulation of energy production, biosynthetic precursors, and redox balance. This metabolic reprogramming serves as a crucial interface between environmental cues and immune cell behavior, shaping both protective immunity and pathological responses. In this section, we summarize the sophisticated metabolic networks governing DC and T cell function, discuss their roles in disease pathogenesis, and highlight how these cutting-edge technologies have transformed fundamental discoveries into potential therapeutic strategies. These technological breakthroughs have not only deepened our mechanistic understanding of immunometabolic regulation but have also revealed novel metabolic checkpoints for clinical intervention.

## Metabolic reprogramming in DC activation and functional maturation

DCs are the most potent antigen-presenting cells and serve as a critical bridge between innate and adaptive immunity. From development to functional maturation, in both homeostasis and disease, immunometabolism plays an essential role in regulating DC generation and function (Møller et al, 2022). Following stimulation through Toll-like receptor (TLR), DCs rapidly reprogram their metabolism, shifting from oxidative

phosphorylation (OXPHOS) to aerobic glycolysis. This transition is driven by the activation of phosphoinositide 3-kinase (PI3K), I kappa B kinase epsilon (IKKε), and TANK binding kinase 1 (TBK1) signalling pathways, along with the translocation of hexokinase 2 (HK2) to the mitochondria (Everts et al, 2014; Krawczyk et al, 2010). This metabolic shift has been shown to be crucial for DC activation and survival. Intriguingly, the elevation of glycolytic activity in DCs occurs in two distinct phases, each regulated by different signaling pathways. In the early hours following TLR stimulation, DCs rapidly mobilize intracellular glycogen stores to support a transient increase in glycolysis (Thwe et al, 2019). This is followed by a sustained phase marked by upregulation of the glucose transporter GLUT1 and increased uptake of extracellular glucose, driven by hypoxia-inducible factor 1α (HIF-1α) and type I interferon (IFN-I) signaling (Pantel et al, 2014). The increase in glycolysis supports citrate metabolism from pyruvate through the p32–pyruvate dehydrogenase pathway, which fuels de novo fatty acid synthesis (FAS), a requirement for DC activation (Gotoh et al, 2018). Fatty acids are further broken down through FAO, generating ATP and key intermediates such as acetyl coenzyme A. These intermediates contribute to broader metabolic and epigenetic regulation (You and Chi, 2023). Glycolysis-driven FAS also promotes the expansion of the endoplasmic reticulum (ER) and Golgi apparatus, which enhances protein synthesis and intracellular transport essential for antigen processing and presentation (Everts et al, 2014; Gotoh et al, 2018). These processes collectively upregulate surface molecules, including costimulatory receptors such as CD40 and CD86, as well as major histocompatibility complex molecules (MHC), which are required for effective T cell activation (Everts et al, 2014). Metabolic reprogramming also influences cytoskeletal remodeling and the oligomerization and surface expression of CCR7, which facilitates DC migration to the lymph nodes, a key step in the initiation of T cell responses (Guak et al, 2018; You and Chi, 2023). In addition to glucose and lipid metabolism, amino acid availability plays a vital role in supporting DC activation, particularly through protein synthesis and support of the TCA. Upon TLR stimulation, human monocyte-derived dendritic cells (moDC) increase the uptake of amino acids such as aspartate, cystine, glutamate, valine, leucine, and isoleucine. The absence of these nutrients in the extracellular environment significantly impairs the expression of costimulatory molecules, cytokine production, and migration (Kakazu et al, 2013). Plasmacytoid dendritic cell (pDC) rely on glutaminolysis for activation and cytokine secretion upon TLR stimulation (Basit et al, 2018). Cytokines including interleukin 3 and granulocyte macrophage colony-stimulating factor (GM-CSF) enhance the expression of neutral amino acid transporters SLC7A5 and SLC3A2, promoting activation of the mammalian target of rapamycin complex (mTORC1). This upregulation increases glycolysis and induces the expression of SLC7A11, contributing to IFN-I and chemokine production (Grzes et al, 2021). Glutamine metabolism is also essential for the effector function of cDC1, particularly in the context of antitumor immunity. Within the TME, competition for glutamine between cancer cells and cDC1 impairs the latter's ability to efficiently cross-present antigens. Mechanistically, glutamine uptake via the SLC38A2 transporter facilitates the assembly of nutrient-sensitive FLCN–FNIP2 complexes, which inhibit the transcription factor TFEB. This regulation of TFEB activity maintains lysosomal homeostasis, thereby

supporting optimal antigen processing and cross-presentation by cDC1 (Guo et al, 2023). Building on these insights, recent studies have employed systems biology approaches such as NetBID, which integrates transcriptomics, proteomics, and phosphoproteomics, have revealed novel mechanisms of mitochondrial-mediated metabolic regulation in DCs. Specifically, the mitochondrial protein Mst1/2, selectively expressed in cDC1, was shown to regulate IL-12 secretion and cross-priming via noncanonical NF-κB signaling—highlighting its role in shaping effector function and T cell responses (Du et al, 2018). Moreover, studies integrating SCENITH with scMEP have revealed distinct patterns of glycolytic reprogramming that govern the differentiation of moDC, along with coordinated activation of mitochondrial pathways during their maturation. Intriguingly, this integrative regulome approach also uncovered a critical role for mTOR and AMP-activated protein kinase (AMPK) signaling, accompanied by the upregulation of OXPHOS, glycolysis, and FAO, collectively promoting the development of tolerogenic moDC phenotypes (Adamik et al, 2022). Together, these findings highlight how glycolysis, lipid, and amino acid metabolism are essential for the metabolic and functional reprogramming of DCs. Disruption of these pathways impairs DC activation, migration, and T cell priming capacity, thereby compromising their central role in orchestrating adaptive immunity. Furthermore, next-generation technologies now enable precise, functional metabolic mapping at the cellular level, enhancing our mechanistic understanding of DC biology and uncovering new regulatory nodes of immune function.

## Metabolic reprogramming and integration in T cells and DCs

Upon receiving antigen-priming, naive T cells undergo profound metabolic transformation to support proliferation, differentiation, and effector function (Chapman et al, 2020). Engagement of the TCR, in concert with costimulatory signals and cytokines, activates the phosphatidylinositol 3-kinase (PI3K)–Akt–mTOR signaling axis, initiating a coordinated metabolic response that includes enhanced glucose uptake and a switch to aerobic glycolysis (Chi, 2012). These bioenergetic demands are paralleled by widespread remodeling of transcriptional networks that govern metabolic gene expression. T cells upregulate multiple metabolic-related transcription factors upon activation to further support their metabolic reprogramming and gene expression. Among the transcription factors upregulated during T cell activation, Myc plays a central role in driving expression of genes involved in glycolysis, such as HK2, that is required for robust glycolysis, and in regulating glutamine metabolism. Myc-dependent metabolic pathways intersect with mTOR signaling to promote glutaminolysis, thereby fueling biosynthetic processes essential for T cell growth and function. In parallel, activated CD8[+] T cells increase their uptake of glutamine and leucine, amino acids critical for sustaining mTOR activity and anabolic metabolism (Nakaya et al, 2014; Sinclair et al, 2013). De novo FAS is also promoted downstream of mTOR signaling following T cell activation (Yang et al, 2013), primarily through the induction of sterol regulatory element-binding proteins (SREBPs) (Kidani et al, 2013). The mTOR–SREBP axis drives the expression of key lipogenic genes, including *Fasn* and *Hmgcr*, facilitating lipid biosynthesis by channeling carbon derived from both glucose and amino acid catabolism. This lipid anabolic

program is essential for supporting membrane biogenesis, proliferation, and effector differentiation. Beyond fulfilling the energetic and biosynthetic demands of activation, metabolically regulated transcriptional and signaling networks play a pivotal role in guiding T cell fate decisions during differentiation, highlighting the intricate interplay between metabolism and immune function. Although HIF-1α is upregulated downstream of the mTOR signaling during T cell activation, its presence is dispensable for early metabolic reprogramming and proliferation. However, HIF-1α–dependent glycolytic programming is critical for directing lineage specification, particularly in modulating the balance between Th17 and regulatory T (Treg) cell differentiation (Shi et al, 2011). While mTOR signaling plays a central role in T cell activation and metabolic reprogramming, its function varies depending on the T cell subset. In effector T cells, mTOR activity is required for differentiation into Th1, Th2, and Th17 lineages, while it concurrently suppresses Treg differentiation (Delgoffe et al, 2009). By contrast, in Treg, mTOR signaling—particularly through mTORC1—is essential for establishing their suppressive function. mTORC1 promotes cholesterol and lipid biosynthesis via the mevalonate pathway, which is necessary for Treg proliferation and their suppressive activity. This mTORC1-mediated metabolic program supports Treg function. Together, these findings highlight the distinct roles of mTOR signaling in different T cell subsets and emphasize the importance of lipid metabolism in maintaining immune homeostasis through Treg (Zeng et al, 2013). Activated CD8+ T cells share several features with DCs in terms of metabolic reprogramming, such as increased glycolysis, glutaminolysis, and FAS. However, while OXPHOS is downregulated in activated DCs, it is increased in CD8$^+$ T cells to support their proliferation and effector function. Following antigen clearance, memory T cells rely on FAO and OXPHOS for long-term survival and recall responses (O'Sullivan et al, 2018; Sena et al, 2013). Metabolic crosstalk also occurs during interactions between DCs and T cells. Although glycolysis is critical for DC activation, excessive glycolytic activity can limit the expression of costimulatory molecules (Amiel et al, 2012). Moderating glycolytic flux may optimize antigen presentation during T cell priming. Moreover, glucose competition between DCs and T cells during immune synapse formation alters DC metabolism by modulating the activity of mTORC1, HIF-1α, and inducible nitric oxide synthase (iNOS), thereby enhancing T cell responses (Lawless et al, 2017). DCs also influence T cell redox metabolism by importing extracellular cystine and converting it to cysteine, which they then release to support T cell proliferation (Angelini et al, 2002). Conversely, tolerogenic DC (tDC) secrete indoleamine 2,3-dioxygenase (IDO), which depletes tryptophan from the microenvironment, limiting T cell proliferation and inducing anergy (Harden and Egilmez, 2012). Together, these findings demonstrate the reciprocal nature of metabolic coordination between DCs and T cells. This exchange of metabolic substrates and the dynamic modulation of metabolic signaling are essential components of immune activation and homeostasis (Fig. 2). However, the details of this reciprocal nature of metabolic coordination remain unclear due to the lack of methodology to delineate the metabolic features at a spatial and cellular level.

### Tumor metabolism

In many pathological settings, including cancer, metabolic constraints imposed by the tissue environment disrupt immune cell function. However, the TME represents an extremely harsh environment to immune cells due to the reduced availability of glucose, oxygen, and amino acids, as well as elevated lactate, acidic pH, and immunosuppressive factors (Kao et al, 2022). These conditions impair DC activation by limiting glycolysis and FAS, leading to decreased cytokine secretion, destabilization of MHC molecules, and impaired antigen presentation (Brown et al, 2020; Caronni et al, 2018; Park et al, 2022). Moreover, increased lipid accumulation within DCs in tumors promotes FAO and OXPHOS, which is associated with the induction of IDO and a tolerogenic phenotype (Gardner et al, 2015; Veglia et al, 2017; Yin et al, 2020). Oxidized lipids have been shown to interfere with the trafficking of MHC molecules, further reducing the ability of DCs to prime T cells (Ramakrishnan et al, 2014; Veglia et al, 2017). In addition, competition for glutamine between DCs and tumor cells impairs the capacity of cDC1 to cross-present antigens and initiate cytotoxic T cell responses (Guo et al, 2023). CD8+ T cells within tumors are exposed to chronic antigen stimulation and nutrient deprivation, leading to exhaustion and diminished effector function (Franco et al, 2020). Limited glucose availability in TME impairs T cell proliferation and cytokine production (Chang et al, 2015; Ho et al, 2015). Mitochondrial dysfunction further contributes to this exhausted state (Yu et al, 2020). Lipid overload in tumor-infiltrating T cells also promotes the expression of checkpoint molecules and exacerbates toxicity through excessive FAO (Ma et al, 2019; Ma et al, 2021). Furthermore, T cells are deprived of key amino acids, including glutamine, arginine, tryptophan, and methionine. They are also exposed to suppressive metabolites such as kynurenine, a breakdown product of tryptophan catabolism by IDO, which promotes T cell dysfunction and cell death (Bian et al, 2020; Geiger et al, 2016; Meireson et al, 2020). These metabolic constraints in the tumor microenvironment impair both DC and T cell function, reducing antitumor immunity. However, these shared features also provide potential therapeutic opportunities. For example, high concentrations of the metabolite D-2HG acts as a well-known oncometabolite that promotes tumorigenesis. Once it is taken up by CD8 T cells, the glycolytic enzyme LDH will directly target it to drive a metabolic program and impair their proliferation, cytotoxicity and interferon-γ signaling with mouse tumor models (Nathan, 2022). Moreover, metabolic interventions such as treatment with L-asparaginase, in combination with immunotherapy or chemotherapy, have been shown to enhance T cell-mediated antitumor responses in cancer patients. This effect is driven by the reprogramming of T cell metabolism and mitochondrial fitness by asparagine deprivation, leading to altered chromatin accessibility and transcription factor networks that enhance their effector function (Chang et al, 2025). These findings underscore the therapeutic potential of targeting the metabolism of immune cells to reinvigorate antitumor immunity. Despite these studies highlighting that DCs and T cells can adapt to metabolic stress in tumors and the adaptation process may orchestrate their differentiation and function, it remains difficult to understand the causality since we do not have a technique which can simultaneously measure metabolic preference, transcriptome and epigenome in the same cells. The development of technology addressing this need may further empower us to unveil how functional adaptation and maladaptation in response to metabolic challenges can tailor immune responses.

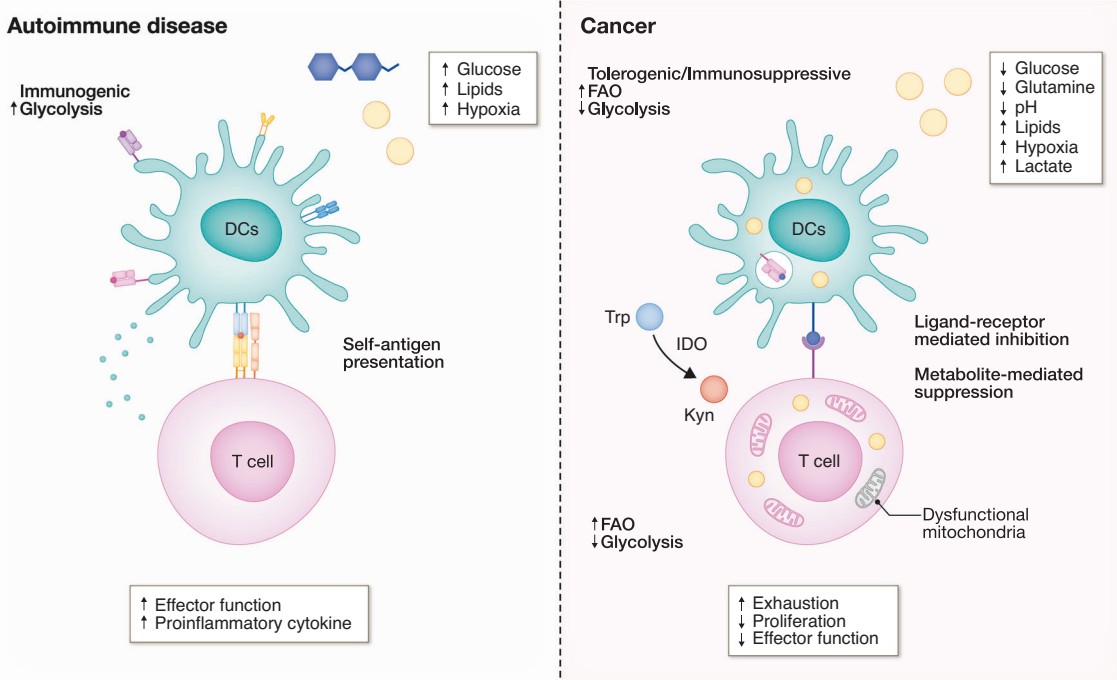

**Figure 2.  Altered metabolic fitness in T cells and dendritic cells modulates their function in disease.**

In TME and autoimmune diseases, the abnormal accumulation or depletion of certain metabolites leads to dysfunction in DCs and T cells. In autoimmune diseases, although specific features vary depending on the disease type, the accumulation of glucose, lipids, and hypoxia collectively drives the activation of DCs. These activated DCs secrete pro-inflammatory cytokines and present self-antigens to T cells, enhancing their differentiation and effector functions, ultimately creating an inflammatory microenvironment that leads to tissue damage. Similarly, in the TME, the depletion of glucose and amino acids, along with the enrichment of lipids, lactate, and hypoxic conditions, impairs DC function. The lack of glycolysis prevents DC activation, priming ability and effector function, while lipid accumulation blocks the translocation of MHC molecules to the cell surface, thereby suppressing their antigen-presenting capacity. Additionally, DCs undergo metabolic reprogramming, increasing FAO to generate energy. This shift promotes a tolerogenic/immunosuppressive DC phenotype, which inhibits T cells either through ligand-receptor interactions or metabolite-mediated suppression. For example, DCs in the TME express IDO, which depletes tryptophan by converting it into kynurenine. Since tryptophan is essential for T cell proliferation, its depletion suppresses T cell function and contributes to immune evasion. Like DCs, T cells in the TME undergo an exhaustion program characterized by impaired mitochondrial function, inhibited glycolysis, and increased reliance on FAO. As a result, exhausted T cells lose their effector functions and proliferative capacity, ultimately weakening antitumor immunity.

## Autoimmune disease and chronic infection

In response to viral infection, DCs undergo metabolic reprogramming similar to that observed following TLR activation. This shift is essential for initiating adaptive immunity and promoting viral clearance. However, as in cancer, chronic infection disrupts DC function. This impairment can result from direct viral infection (Baca Jones et al, 2014) or from immunosuppressive environmental cues that rewire DCs toward a tolerogenic phenotype. Recent studies indicate that metabolic insufficiency is a major contributor to DC dysfunction in chronic infections. pDC from virus-infected individuals exhibit defects in both OXPHOS and glycolysis, leading to reduced IFN-I production (Greene et al, 2025). These findings underscore the importance of metabolic reprogramming in shaping DC function during chronic infection. Although the precise mechanisms that control DC dysfunction and tDC differentiation in chronic infection remain poorly defined, insights can be drawn from the study of tDC in the tumor microenvironment. Compared to immunogenic DCs, tDC display unique metabolic profiles that support their suppressive behavior. Interestingly, not all chronic inflammatory diseases suppress DC immunogenicity in the same way as cancer or persistent viral infections. In autoimmune conditions such as rheumatoid arthritis (RA), systemic lupus

erythematosus (SLE), and diabetes, DCs often exhibit enhanced activity, which exacerbates disease progression through increased antigen presentation and pro-inflammatory cytokine secretion. For instance, DCs from RA patients demonstrate elevated glycolysis, enhanced expression of costimulatory molecules, and increased production of inflammatory cytokines relative to healthy controls. This may result from hypoxia-induced glycolytic activation within inflamed tissues, which sustains DC immunogenicity (Coutant and Miossec, 2016). Similar metabolic features are observed in diabetes, where increased glucose and lipid levels, together with local hypoxia, promote a pro-inflammatory phenotype in DCs. Although direct causal links between DC metabolism and disease progression in diabetes remain to be fully established, DCs exposed to high-glucose and fatty acid-rich conditions skew T cell differentiation toward T helper 1 (Th1) and phenotypes (Brombacher and Everts, 2020; Mbongue et al, 2017). Together, these findings suggest that while nutrient deprivation impairs DC function in cancer and chronic infections, nutrient excess may promote hyperactive DC responses in autoimmunity. This dual role of DCs as either promoters or suppressors of immune activation highlights the therapeutic potential of targeting DC metabolism to modulate immune responses in a disease-specific context. In chronic viral

infection, prolonged antigen exposure, sustained inhibitory receptor signaling, and a dysregulated microenvironment drive T cell dysfunction. This culminates in a progressive loss of effector function and the development of an exhausted phenotype. Exhausted T cells are metabolically impaired, exhibiting defective glycolysis and mitochondrial dysfunction (Schurich et al, 2016). Additionally, chronic stimulation induces stable epigenetic changes, referred to as epigenetic scars, which persist even after antigen clearance and contribute to long-term functional impairment (Yates et al, 2021). These changes in transcriptional networks, metabolic pathways, and epigenetic regulation reinforce the exhaustion program and limit immune reactivation (Ma et al, 2025). Despite these challenges, metabolic interventions have shown promise in restoring T cell function. Strategies such as overexpression of the mitochondrial regulator PGC-1 alpha (PGC-1α), administration of interleukin 12, or antioxidant treatment have been reported to improve mitochondrial health, rescue effector functions, and promote the development of stem-like T cell states (Bengsch et al, 2016; Schurich et al, 2016; Vardhana et al, 2020). Interestingly, features of T cell exhaustion are not restricted to cancer and infection. Similar metabolic and transcriptional patterns have been observed in autoimmune diseases such as SLE, diabetes, and RA (Gao et al, 2022). Although the drivers of T cell exhaustion in these settings are not fully understood and may vary between diseases, nutrient availability, hypoxia, and lipid accumulation appear to be recurring factors. Importantly, while reversing T cell exhaustion may enhance immunity in cancer and infection, inducing or maintaining exhaustion could be beneficial in autoimmune contexts by tempering self-reactivity (Sharabi and Tsokos, 2020; Weyand et al, 2020; Zhang et al, 2022). These observations reinforce the central role of metabolism in regulating T cell fate and function across diverse diseases. Modulating metabolic pathways offers a promising strategy to fine-tune T cell responses in a context-dependent manner, with applications in both immune activation and immune tolerance.

## Bottlenecks and challenges in studying DC and T cell metabolism

The central role of immunometabolism in regulating DCs and CD8+ T cell behavior is now well recognized, both in homeostasis and disease. However, dissecting how metabolic pathways and intercellular crosstalk are coordinated in vivo and in situ remains technically and conceptually challenging. These obstacles stem from both limitations in current experimental models and the inherent complexity of immune cell biology. In the following sections, we highlight major unresolved questions and technical bottlenecks that constrain progress in the field.

### Limited in vitro models for studying DC metabolism
Despite considerable progress in understanding DC biology, studying DC metabolism remains hindered by several major constraints. DCs are rare in vivo and notoriously difficult to maintain in vitro, limiting their accessibility for mechanistic and metabolic investigation (Vremec et al, 2015). Moreover, a lack of genetic mouse models that allow selective targeting of specific DC subsets further complicates in-depth study. Consequently, much of our current knowledge is derived from in vitro systems, particularly cultures of bone marrow-derived DCs (BMDCs). Two principal

cytokines are employed to generate BMDCs in mice: GM-CSF and FMS-like tyrosine kinase 3 ligand (Flt3L). Flt3L promotes the development of all major DC subsets both in vitro and in vivo (Karsunky et al, 2003), with Flt3L-cultured BMDCs producing a heterogeneous mix of cDC1, conventional dendritic cell 2 (cDC2), and pDC, thereby more closely mimicking the native DC repertoire (Lutz et al, 2023). These cultures can be further manipulated to enrich specific populations. For instance, GM-CSF enhances CD103 + DC differentiation (Mayer et al, 2014), while co-culture with OP9-DL1 stromal cells expressing Notch ligands facilitates development of CD8α + cDC1 (Kirkling et al, 2018). Alternatively, immortalized progenitor cells engineered to express an estrogen receptor–HoxB8 fusion protein offer a scalable platform for generating DCs (Redecke et al, 2013). This approach is particularly suitable for high-throughput applications such as CRISPR screening and allows for genetic manipulation of DCs. Although Flt3L-based cultures generate physiologically relevant DC subsets, purification is still required for functional and metabolic analyses. GM-CSF-based cultures, with or without interleukin 4 or lipopolysaccharide, tend to produce inflammatory DCs alongside macrophage-like cells. These GM-CSF-derived DCs are competent in antigen presentation but display a more inflammatory phenotype even in the absence of stimulation (Helft et al, 2015). Their heterogeneous composition and activation state may not accurately reflect native DC biology. Thus, while current in vitro models are informative, they do not fully recapitulate the complexity of DCs in vivo. Concerns regarding the survival, transcriptional similarity, and functional relevance of cultured DCs remain, and these limitations contribute to the variability observed across different studies. Further improvements in culture conditions and the development of tools for metabolic analysis of rare immune cells are necessary to overcome these obstacles.

### Subset-specific metabolic preferences between DCs and CD8 T cells
DCs include multiple subsets, including cDC1, cDC2, pDC, and moDC, each shaped by tissue environment and inflammatory cues. Considerable heterogeneity also exists within these subsets. For example, DCs found in tumors, intestinal tissues, and lymphoid organs can show distinct transcriptional programs and functional specializations. scRNA-seq has revealed additional populations such as DC3 (Bourdely et al, 2020), mature DCs enriched in immunoregulatory molecules, and CCR7-expressing migratory DCs (Lee et al, 2024), which range from immunostimulatory to immunosuppressive in function. Despite these features, many of these subsets remain poorly defined, and their metabolic characteristics are still unknown. Understanding how metabolism shapes subset-specific function is essential for guiding the development of DC-based therapeutic strategies. A similar issue arises in the study of CD8+ T cells, particularly in exhausted T cell subsets that are central to chronic infection and cancer. Exhausted T cells follow a progressive differentiation program with progenitor, intermediate, and terminal stages, each associated with unique metabolic signatures (Beltra et al, 2020; Franco et al, 2020). Metabolic remodeling influences these stages by altering transcriptional and epigenetic landscapes (Franco et al, 2020), but it remains unclear whether these metabolic changes drive exhaustion or occur consequently. Dissecting how metabolic changes correlate with T cell differentiation may reveal opportunities to maintain progenitor-like features or restore effector function. Approaches

that modulate metabolism could provide new therapeutic tools to reshape T cell responses in disease (Schurich et al, 2016; Wu et al, 2023).

### Challenges in studying DC and T Cell metabolism in diseases and their metabolic crosstalk

In disease settings, immune cell metabolism is influenced by various environmental factors, including cytokines, metabolites, and interactions with other immune cells (Kao et al, 2022). However, current tools are limited in their ability to capture the full complexity of these interactions. This lack of resolution makes it difficult to pinpoint mechanisms of dysfunction and slows the translation of metabolic insights into therapy. Lipid accumulation, for example, has been implicated in DC dysfunction (Bougneres et al, 2009; Ramakrishnan et al, 2014), but many questions remain unresolved. It is still unclear which cell types contribute to lipid accumulation, which lipid species are responsible for impairing function, and whether these effects are consistent across all DC subsets. Certain lipids may play both activating and suppressive roles in different DC contexts (You and Chi, 2023), suggesting the need for subset-specific metabolic profiling. Many recently identified DC subsets also remain poorly characterized in disease. Their metabolic requirements and immunological roles are largely unknown. Broad targeting of metabolic pathways could inadvertently suppress both immunostimulatory and regulatory DCs, reducing therapeutic benefit. Similarly, T cell exhaustion involves significant metabolic dysregulation, yet the precise pathways involved have not been fully defined. In vivo, T cell function is shaped by dynamic interactions with DCs, spatial positioning within tissues, and signals from surrounding cells. These complex interactions are difficult to replicate in conventional in vitro systems, which limits the ability to study metabolic crosstalk during immune responses.

### Technical limitations for dissecting metabolism in DCs and T cells

One major limitation is that although multi-platform approaches now enable the simultaneous investigation of metabolic preferences alongside other regulatory layers, such as transcriptomics and proteomics, no existing method can concurrently assess metabolism, gene expression, epigenetic modifications, and chromatin remodeling within the same single cell. While independent analyses of these layers have provided valuable insights, integrating them remains challenging. Several factors complicate this integration, including cellular heterogeneity, experimental variability, and discrepancies between mRNA and protein expression levels. Moreover, the complexity of metabolite networks makes it difficult to draw precise conclusions from separate studies. The inability to capture all regulatory layers simultaneously limits our understanding of how metabolic activity is coordinated with gene expression and chromatin remodeling within individual cells. Another key challenge is our limited understanding of the metabolic rewiring that occurs during interactions between DCs and T cells, which is essential for effective T cell activation. Although genetic tools and metabolic profiling offer some insight, they are often indirect and fail to capture the full dynamics of these interactions. Most current studies rely on endpoint measurements of T cell activation and function, providing only a partial view of the processes involved. It remains unclear how metabolic reprogramming during DC–T cell interactions initiates and

supports transcriptomic changes and chromatin remodeling necessary for full T cell activation. This knowledge gap highlights the absence of adequate tools to study the metabolic aspects of cell–cell communication and to integrate multi-layered molecular information. To overcome these limitations, future studies should incorporate real-time imaging, spatially resolved technologies, and single-cell multi-omics approaches. These should include the integration of metabolomics, transcriptomics, epigenetic profiling, and chromatin accessibility analyses at the single-cell level. Comprehensive approaches of this nature will be crucial in uncovering how metabolism governs not only individual immune cell states but also their interactions within complex tissue environments. Moreover, combining single-cell technologies, live imaging, metabolomic profiling, and spatial transcriptomics will provide deeper insights into the metabolic regulation of immune cells. Addressing these technical challenges will require the development of innovative experimental platforms, advanced computational algorithms, and the application of artificial intelligence to manage and interpret high-dimensional datasets. Successfully overcoming these obstacles will provide a more nuanced understanding of how metabolic regulation operates across multiple layers of cellular control and intercellular communication, ultimately shedding light on how metabolic preferences influence immune function in health and disease.

# Emerging therapeutic interventions targeting immunometabolism: harnessing rapidly advancing metabolic approaches

This section reviews the current technical limitations in dissecting metabolism at the single-cell and tissue level, explores how emerging insights into immunometabolism are being translated into DC-based therapeutic strategies, and highlights future directions driven by cutting-edge technological innovations. Together, these developments pave the way for the rational design of metabolism-targeted immunotherapies.

## Ongoing and future perspectives on DC-based metabolic therapy

Recent advances in immunometabolism have demonstrated that metabolic reprogramming plays an important role in DC activation, antigen presentation, and T cell priming. DC-based therapies are being explored as a potential approach in immunotherapy due to their unique ability to influence T cell responses. While DC-based therapies are not yet widely established for autoimmune conditions or chronic viral infections, encouraging clinical outcomes have been reported in certain contexts, such as prostate cancer (Kantoff et al, 2010). These findings support further investigation into the therapeutic potential of DCs. While a comprehensive review of all DC-based therapy approaches is beyond the scope of this review, as these have been extensively discussed elsewhere (Passeri et al, 2021; Shbeer, 2024), several strategies have been developed to target and manipulate the function of DCs, either ex vivo or in vivo. These approaches include: (1) Adaptive cell therapy by transferring ex vivo-generated manipulated DCs; (2) In vivo targeting of DCs using antibodies (such as Clec9a and CD40); (3) DC vaccination by administration

of lipid vesicles or nanoparticles that contain a stimulus to prime DCs in vivo; (4) Combining DC therapy with radiotherapy, chemotherapy, and immunotherapy. These strategies have been evaluated in multiple ongoing and completed clinical trials for DC-based therapies across various diseases (Table 2). Collectively, these studies underscore the potential of DC-based therapies in cancer, infectious diseases, and autoimmune conditions. Given their capacity to mediate both immunogenic and immunoregulatory effects, DCs may offer versatility in therapeutic applications. However, challenges remain in translating this concept into clinical success. One notable hurdle is the limited migration of transferred DCs to lymph nodes, which is critical for effective T cell activation. Low migratory efficiency and cell viability limit the effectiveness of DC-based therapies and restrict their routes of administration. This may be linked to metabolic constraints, as early DC activation relies on glycolysis, which supports CCR7 expression and migratory capacity (Guak et al, 2018). Further investigation into these mechanisms or targeting glycolysis in DCs could be one way to enhance DC trafficking in therapeutic contexts. Another key consideration is the type of DC used. Many clinical trials have utilized moDCs, yet accumulating evidence suggests that DCs derived from CD34[+] precursors or directly isolated from blood may retain superior functionality and subset-specific specialization (Garg et al, 2017; Wculek et al, 2020). These alternative sources could potentially be tailored for disease-specific applications. Since metabolic programming is closely linked to DC differentiation (Pearce and Everts, 2015), manipulating metabolic pathways during DC generation and maturation may provide opportunities to direct differentiation toward desired subsets. This approach could offer a strategy for generating functionally optimized DCs with improved therapeutic potential. Third, metabolic stress is a key factor that alters cell behavior in various diseases. Reprogramming DC metabolism represents a possible avenue to counteract the immunosuppressive and nutrient-limited tumor microenvironment as well as nutrient-enriched autoimmune diseases. By manipulating DC metabolism either ex vivo before adaptive transfer or directly targeting DCs in vivo, it may help sustain their function and improve immune responses under metabolically challenging conditions. Emerging findings in immunometabolism continue to shed light on how metabolic cues shape DC behavior, suggesting that targeting metabolism could complement existing therapeutic strategies. Although preclinical mouse models have shown promising outcomes in enhancing antitumor responses via metabolic modulation of DCs (Inamdar et al, 2023), clinical translation remains at an early stage. Further investigation into the metabolic regulation of DCs will be important for overcoming current limitations and advancing therapeutic applications. Lastly, the capacity of DCs to influence T cell metabolism represents an intriguing and underexplored mechanism for immune modulation. This approach goes beyond conventional ligand-receptor interactions by directly manipulating T cell behavior through metabolic reprogramming.

## Metabolic reprogramming of T Cells: toward translational immunotherapy

While the application of DC-based therapies remains relatively limited, there is growing interest in targeting T cell metabolism as a strategy to enhance immune responses in cancer and other diseases.

This concept has gained attention in efforts to improve the efficacy of CAR-T cell therapies. By engineering T cells to alter their metabolic activity, researchers have been able to enhance their functional capacity and resistance to the nutrient-deprived conditions characteristic of the TME. For example, one study demonstrated that overexpressing the fructose transporter GLUT5 in CAR-T cells enabled them to efficiently utilize fructose as an alternative carbon source in glucose-limited environments, thereby boosting their antitumor activity (Schild et al, 2025). This metabolic modification could be further augmented by providing a fructose-enriched diet, enhancing T cell responses in vivo. Similarly, another study modulated T cell metabolism by treating cells with dichloroacetate (DCA) during in vitro expansion. DCA inhibits pyruvate dehydrogenase kinase 1 (PDHK1), a key regulator that promotes glycolysis during T cell activation. Inhibiting PDHK1 shifts T cell metabolism toward mitochondrial oxidative phosphorylation. Although DCA-treated CAR-T cells did not show enhanced immediate effector function, they exhibited improved infiltration and survival within the TME, along with reduced terminal exhaustion and an increased gene signature of progenitor-like cells (Frisch et al, 2025). These effects may be linked to acetyl-CoA-mediated histone modifications driven by enhanced mitochondrial metabolism. Metabolic reprogramming strategies such as these offer promising avenues to overcome key limitations of current CAR-T therapies, particularly in solid tumors where T cell function is impaired by nutrient competition and metabolic stress. Recent findings show that overexpression of a hyperactive form of PGC-1α in CAR-T cells, which promotes mitochondrial biogenesis, can further enhance antitumor immunity (Lontos et al, 2023). Together, these findings suggest that fine-tuning T cell metabolic programs can significantly enhance the durability and effectiveness of adoptive cell therapies, especially in metabolically challenging TME. Beyond cancer, T cells also display remarkable functional plasticity as DCs, with roles that extend from promoting inflammation to mediating immune tolerance. This dual potential positions T cells as therapeutic targets not only for cancer, but also for autoimmune diseases. Studies have explored the possibility of directing T cells toward regulatory phenotypes by modulating key metabolic pathways. For instance, inhibition of methylenetetrahydrofolate dehydrogenase 2 (MTHFD2), an enzyme involved in purine biosynthesis, increases FOXP3 expression and promotes Treg differentiation. This occurs via mTORC1 inhibition and is accompanied by reduced Th17 cell proliferation, leading to therapeutic benefits in an Experimental Autoimmune Encephalomyelitis (EAE) model (Sugiura et al, 2022a). In another example, the clinically approved drug metformin has been shown to activate AMPK and suppress mTOR signaling, thereby reducing Th17-driven inflammation while enhancing Treg differentiation in models of inflammatory bowel disease (IBD). Metformin also inhibits STAT3 activation, a key transcription factor in Th17-mediated inflammation (Lee et al, 2015). These metabolic modulations not only reduce pathogenic T cell responses but also support the expansion of regulatory populations that maintain immune homeostasis. Collectively, these approaches underscore the therapeutic promise of metabolic modulation in T cells. Whether enhancing effector function in cancer or promoting immune tolerance in autoimmune diseases, metabolic rewiring offers a promising strategy to tailor T cell behavior—either by generating desired T cell subsets in vitro for adoptive cell therapy or by directly

**Table 2. Overview of clinical trials on DC-based immunotherapies for cancer and autoimmune diseases.**

| Clinical trial ID | Disease types | DC source | Treatment | Phase | Status | Clinical outcomes |
|---|---|---|---|---|---|---|
| NCT06152367 | Melanoma | IL-4 and GM-CSF- derived moDC | Loaded with melanoma cell line antigen and a low dose of IL-2 | Phase II | Completed | Overall survival; Progression-free survival; Systemic cytokine measurements; Systemic cytotoxic T cell-mediated antitumor cell cytotoxicity |
| NCT03782064 | Multiple myeloma | DC/myeloma fusion cells | Combine with nivolumab | Phase II | Terminated | Unspecified |
| NCT05650918 | Pancreatic cancer | moDC | Combine with CD40 agonists | Phase I | Completed | T cell influx; Immune cell subsets; Immuno-oncology gene expression signatures |
| NCT04911621 | Glioma | moDC | TAA mRNA-loaded DCs plus temozolomide-based chemoradiotherapy | Phase II | Active | Overall survival; Progression-free survival; Tumor-reactive CD8 T cell numbers and function |
| NCT03119025 | Chronic HCV-infection | IFN-α and GM-CSF- derived moDC | Pulsed with HCV core and NS3 antigens | Phase II | Completed | HCV viral load; T cell proliferation and cytotoxicity |
| NCT02767193 | HIV | moDC | Pulsed with inactivated HIV virus | Phase I | Completed | HIV viral load; T cell proliferation and cytotoxicity; DC maturation marker expression; Autoimmunity and inflammatory biomarker; microbial translocation |
| NCT00445913 | Type 1 diabetes | Immunoregulatory moDC | Antisense oligonucleotides that target CD40, CD86, and CD80 | Phase I | Completed | Unspecified |
| NCT04590872 | Type 1 diabetes | Tolerogenic DC | With proinsulin peptides | Phase I | Active | Number of monocytes and DC; C-peptide area (insulin production); IFN-γ and IL-10 secreting CD4 T cell; Autoreactive CD8 T cell population; Immune population; Islet autoantibodies concentration; glucose level in blood |
| NCT00510133 | AML | moDC | Transfect with mRNA that encodes TAAs | Phase II | Completed | Unspecified |
| NCT05834296 | Alzheimer's dementia | IL-4 and GM-CSF-derived moDC | Pulsed with the amyloid beta mutant peptide | Phase II | Active | anti-Aβ antibodies; Patient blood pressure / heart rate / oxygen saturation |
| NCT02618902 | Multiple sclerosis | Tolerogenic DC | Adaptive cell transfer | Phase I | Completed | Disability status; Blood lymphocyte phenotyping and cytokine profiling; Myelin-reactive T cell reactivity |

targeting metabolic signaling pathways to influence T cell function and differentiation in vivo.

## Outlook and future directions

To develop effective metabolic interventions for disease, it is essential to overcome current technical challenges and achieve a comprehensive understanding of metabolic pathways at both the single-cell level and within complex tissue environments. The rapid advancement of technologies such as single-cell and spatial multi-omics technologies (Semba and Ishimoto, 2024), CRISPR-based metabolic screens, multi-dimensional metabolic profiling (Ganesh et al, 2021), and advanced imaging mass spectrometry has dramatically expanded our capacity to investigate cellular metabolism with unprecedented resolution. These innovations now enable the development of functional therapies that target immune cell metabolism, offering promising avenues for treating cancer, autoimmune disorders, and infectious diseases. Looking ahead, the integration of artificial intelligence and machine learning with these high-dimensional datasets will further enhance our ability to model, predict, and manipulate metabolic networks in real time. Importantly, these tools also open the possibility of understanding how genetically identical T cells—such as those sharing a T cell receptor targeting the same neoantigen—can exhibit divergent behaviors depending on the metabolic constraints they encounter in different regions of a tumor. This level of insight will be critical for advancing precision immunotherapies. Ultimately, the convergence of next-generation technologies, computational power, and biological insight is poised to transform immunometabolism research. Translating these discoveries into clinical applications will require interdisciplinary efforts but holds the potential to reshape our approach to immune-mediated diseases and usher in a new era of targeted metabolic therapies.

## Peer review information

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

## Acknowledgements

P-CH is supported in part by the Ludwig Cancer Research Immunometabolism initiative, the Swiss Cancer Foundation (KLS-5949-08-2023), Swiss National Science Foundation (310030_215126, 310030L_208130, TMCG-3_213736, and IZLCZ0_206083, CRSII5_205930), the Cancer Research Institute (Lloyd J. Old STAR award), and the Helmut Horten Stiftung. Figures in this work were created using BioRender.

## Author contributions

**Yi-Hao Wang**: Writing—original draft; Writing—review and editing. **Limei Wang**: Writing—original draft; Writing—review and editing. **Ping-Chih Ho**: Conceptualization; Writing—original draft; Writing—review and editing.

## Disclosure and competing interests statement

P-CH is a co-founder of Pilatus Biosciences.

