## [Peer Review File · The EMBO Journal]

Decoding immunometabolism with next-generation tools: lessons from dendritic cells and T cells

Yi-Hao Wang, Limei Wang, and Ping Chih Ho

Corresponding authors: Ping Chih Ho (ping-chih.ho@unil.ch) , Limei Wang (limei.wang@unil.ch)

Review Timeline:

Submission Date:	17th May 25
Editorial Decision:	5th Jun 25
Revision Received:	23rd Jul 25
Editorial Decision:	31st Jul 25
Revision Received:	6th Aug 25
Accepted:	4th Sep 25

Editor: Daniel Klimmeck

Transaction Report:

Dear Dr Ho,

Thank you again for sending us your review article manuscript for consideration by the EMBO Journal. As mentioned, we have asked two dedicated experts in cancer immunology and metabolism to assess your manuscript, and we have received feedback from both of them, which I enclose below.

As you will see, the experts much appreciate the review and find it timely and worth publishing. They also provide constructive feedback on how to further improve it by advancing the discussion, as well as clarifying a number of aspects. In particular, they encourage you to broaden the scope of the piece by integrating analyses on macrophages and consequences of metabolic immune crosstalk for the TME. They also suggest including additional information on methodologies suitable for studying immune cell metabolism. Further, the experts make suggestions to complement the literature references cited.

I hope you will find the comments helpful. I am sure that an amended version incorporating the suggestions made by the referees will be highly noted and appreciated. I agree that an 'Outlook' section will be very helpful to the readers. Thus, I would like to invite you to submit such a revised version using the link enclosed below.

Please let me know any time in case I can be of any help with this.

with
Best wishes,

Daniel

Daniel Klimmeck, PhD
Senior Editor
The EMBO Journal

Referee #1:

The manuscript by Wand YH et al. summarizes the tools and technologies to investigate the metabolic regulation of immune system and the immunometabolism in both healthy and disease contexts. Accumulated evidence indicates that metabolic reprogramming orchestrates the functionality of immune cells including T cells, macrophages and DCs. Dysregulated cellular metabolic programs are associated with functional alternations of these cells in multiple diseases such as autoimmune diseases and cancers. Exploring metabolic profiling in immune cells will provide the new insights for mechanisms linking metabolism and immune signaling, and targeting such metabolic pathways is a potential means to treat immune-related disease. Overall, this is a well-written, informative, and insightful review that will benefit readers and researchers not only in the embolism field but also in immunology. Several additional points need to be discussed before acceptance for publication.

1. The core of the review aims to discuss the advanced tools and approaches for dissecting metabolism in immune system and related diseases. However, most discussion is limited in the biological aspects of metabolic regulation in immune response. More detailed, comprehensive and structured summary and discussion of metabolic tools and technologies are needed to be incorporated accordingly. For example, a table or figure to summarize the current technologies would be helpful, as well as the challenges and limitations of using these tools.
2. Instead of T cell and DCs, macrophage is the most studied cells in immunometabolism field. The authors can consider adding some discussion regarding the applications of current metabolic approaches in this cell.
3. Many discussions such as DC-T cell crosstalk remain descriptive. More detailed and mechanistic conclusion and/or

perspective are needed to be provided.

4. In section 3, the authors discussed the perspective of metabolic reprogramming to orchestrate DC function in disease control. However, there are very limited literatures visiting this topic. Discussion of this point in a Review paper may cause the impression that manipulation of DC metabolism is a well-adopted approach. Instead, reprogramming of T cell metabolism to achieve immunotherapy in cancers and autoimmune diseases is more explored (PMIDs: 39879981; 36914208; 40020672). Tune-down of metabolism and DC therapy but include T cell metabolism in immunotherapy is needed for an updated manuscript.

5. More detailed metabolic regulations in DC biology such as <https://pubmed.ncbi.nlm.nih.gov/29849151/> and <https://www.nature.com/articles/s41467-022-32849-1> need to be included.

6. Discussions regarding metabolism in T cell regulation can be improved. The interaction between metabolic programs and immune pathways, as well as signaling (<https://pubmed.ncbi.nlm.nih.gov/21708926/>; <https://pubmed.ncbi.nlm.nih.gov/22195744/> <https://pubmed.ncbi.nlm.nih.gov/23812589/>) should be incorporated into the manuscript accordingly.

Referee #2:

This manuscript presents a comprehensive and well-organized review of how cellular metabolism governs the effector functions and fate decisions of immune cells, with particular emphasis on dendritic cells (DCs) and T cells in both physiological and pathological contexts. The authors effectively summarize recent advances in immunometabolism, emphasizing how emerging technologies, such as CyTOF, SCENITH, BONCAT, single-cell metabolic profiling (scMEP), in vivo CRISPR screening, and integrated multi-omics have transformed our understanding of immune cell metabolism. Notably, the shift from bulk metabolic analyses to high-resolution single-cell and spatial approaches has enabled precise characterization of immune cell metabolic states and opened new avenues for therapeutic modulation of immune responses in complex disease settings, including cancer, autoimmunity, and infection. Overall, this review is timely and well-crafted, and it will serve as a valuable resource for both immunologists and translational researchers, provided that the following information is included.

1. The discussion of metabolic crosstalk between DCs and T cells-such as competition for glucose and amino acids-is intriguing but limited. Expanding on how these interactions are shaped by and contribute to distinct disease microenvironments would add important depth.
2. Including a brief "Outlook" or "Future Directions" section summarizing key challenges and opportunities for translating insights in immunometabolism into clinical therapies would enhance the manuscript's impact and provide a forward-looking perspective.
3. Table 1 would benefit from additional details, such as trial phase and status, key clinical outcomes, and clarification on whether metabolic modulation was a primary therapeutic strategy.

The authors addressed the requested issues.

Dear Dr Ho,

Thank you and your colleagues for resubmitting the revised review article manuscript, as well as for your patience with our feedback. As indicated earlier, we have asked both experts to reassess your amended manuscript version. We have received additional comments from referee #1, which I enclose below. Please note that while referee #2 was at this point not able to look back into the work, I have assessed your response to his-her points and found them to be very well addressed.

I am thus pleased to let you know that your review article has been provisionally accepted for publication at the EMBO Journal.

I still need you to consider a number of changes and amendments with respect to formatting of the manuscript as indicated below. I also went again through the text and share an edited version attached FYI. Please let me know if these changes-amendments would be fine with you. Also, should you have questions related.

For the figures, I think we can move forward with these versions pending minor adjustments (see below).

I am looking forward to your feedback on this,
and seeing the piece accepted and at production shortly.

Best wishes,

Daniel Klimmeck

Daniel Klimmeck, PhD
Senior Editor
The EMBO Journal

EMBOJ-2025-121394, final formatting adjustments required:

- >> Please adjust the reference mode to EMBO Journal style.
- >> add up to five keywords to the manuscript.
- >> Acknowledgements, Funding: Swiss Cancer Foundation and Helmut Horten Stiftung should be added in our online system, and the project numbers should be added where available.
- >> Competing Interests' should be renamed into 'Disclosure and competing interests statement'.

- >>Figures: please remove the figures from manuscript text and provide as separate high-resolution image files.
- >>Figure 1: please correct the typo 'Searhorse'

- >>Figure legends: should be compiled at the end of the manuscript text.
- >> The list of abbreviations should be removed.
- >> Table 1 needs a short legend added

Referee #1:

The authors have done an excellent job. I am supportive of the manuscript.

The authors addressed the remaining formatting changes.

Dear Dr Ho,

Thank you for submitting the revised version of your review manuscript for consideration by the EMBO Journal.

I have carefully checked your amendments towards the experts' comments and found them to be well addressed and plausibly integrated into the revised text. I am thus very pleased to inform you that your review article has now been accepted for publication in the EMBO Journal.

As mentioned, I have already passed the figures on to our graphics editorial team, who will approach you shortly regarding the versions translated into our journal style.

Your manuscript will be processed for publication by EMBO Press. It will be copy edited and you will receive page proofs prior to publication. Also, you will soon be contacted by Springer Nature to sign your publishing license.

Should you experience any difficulty, please email publishing@embo.org.

If you have any questions, please do not hesitate to contact me or the Editorial Office.

I look forward to progressing swiftly towards online publication of this review article!

with
Best wishes to Lausanne,

Daniel Klimmeck

Daniel Klimmeck, PhD
Senior Editor
The EMBO Journal
EMBO
Postfach 1022-40
Meyerhofstrasse 1

D-69117 Heidelberg
contact@embojournal.org
Submit at: <http://emboj.msubmit.net>